# Strong in-plane scattering of acoustic graphene plasmons by surface atomic steps

Ni Zhang[1,4], Weiwei Luo[1,4], Lei Wang[2,4], Jiang Fan[1], Wei Wu[1], Mengxin Ren[1], Xinzheng Zhang [1], Wei Cai [1,3 ✉] & Jingjun Xu [1 ✉]

Acoustic graphene plasmons (AGPs) have ultrastrong field confinement and low loss, which have been applied for quantum effect exploration and ångström-thick material sensing. However, the exploration of in-plane scattering of AGPs is still lacking, although it is essential for the manipulation of ultraconfined optical fields down to atomic level. Here, by using scattering-type scanning near-field optical microscopy (s-SNOM), we show that the mid-infrared AGPs can be strongly scattered by atomic level height steps, even though the step height of the scatterer is four orders of magnitude smaller than the incident free wavelength. This effect can be attributed to larger back scattering of AGPs than that of the traditional graphene plasmons. Besides, the scattering of AGPs by individual scatterers can be controlled via electrical back gating. Our work suggests a feasible way to control confined optical fields with atomic level height nanostructures, which can be used for ultra-compacted strong light–matter interactions.

---

[1] The Key Laboratory of Weak-Light Nonlinear Photonics, Ministry of Education, School of Physics and TEDA Institute of Applied Physics, Nankai University, Tianjin 300457, China. [2] College of Physics and Electronic Engineering, Xinyang Normal University, Xinyang 464000, China. [3] Collaborative Innovation Center of Extreme Optics, Shanxi University, Taiyuan, Shanxi 030006, China. [4] These authors contributed equally: Ni Zhang, Weiwei Luo, Lei Wang. ✉email: weicai@nankai.edu.cn; jjxu@nankai.edu.cn

Graphene plasmons (GPs)-electromagnetic fields coupled to charge carrier oscillations, have attracted a great deal of attention owing to their short wavelengths, strong field confinement, and electrical tunability[1–4]. These unique properties make GPs potential applications for controlling electromagnetic waves on the nanometer scale[5], such as highly integrated sensitive spectroscopy[6], modulators[7], and detectors[8–10]. Moreover, by placing the graphene sheet close to a metallic surface, the GPs hybridize with their mirror image, leading to an electromagnetic mode comprising anti-phase charge oscillations in the graphene and the metal[11]. In contrast to GPs, whose energy scales with the square root of their momentum, this new hybridized mode satisfies linear energy versus momentum dispersion, which is named acoustic graphene plasmons (AGPs)[12–14]. AGPs lead to ultra-strong confined and enhanced electric fields inside the gap between the two materials, which can be confined in-plane extensively to almost 1/300 of their equivalent free-space wavelength[15]. In further, by taking advantage of the plasmon confinement down to the length scale of atomic level, quantum effects have been explored with AGPs[16,17]. Besides, the field concentration provided by AGPs leads to improved sensitivity of infrared molecular vibrational spectroscopy down to ångström-thick material layers[11,18]. In addition, AGPs can boost nonlinear processes, such as second-harmonic generation or four-wave mixing[19], providing a platform to explore enhanced nonlinear light–matter interactions.

Yet, though lots of work on AGPs have been reported, the experimental progress is still few. Specifically, both near-field photocurrent nonoscopy[14,16] and far-field infrared spectrum[11,15,17] are used to explore the properties of AGPs in either THz or infrared ranges, however, the direct detection of AGPs through infrared near-field optical scattering[20] is still difficult, especial on the in-plane scattering properties of AGPs. And this is essential for the manipulation and utilization of ultraconfined optical fields down to atomic level, where the widely applications of two-dimensional atomic height cavities are booming thanks to the advancement of two-dimensional van der Waals materials[21]. Here, relying on scattering-type scanning near-field optical microscopy (s-SNOM), mid-infrared AGPs modes in h-BN encapsulated graphene deposited on top of Au surface with atomic level thickness steps were imaged and analyzed. Because of the strong confined electromagnetic fields of AGPs, strong plasmon scattering is observed despite the step height is as low as 3 nm, which is about four orders of magnitude smaller than the incident-free space wavelength. Moreover, the distribution of near-fields for AGPs can be efficiently tuned in situ by gating the graphene. Our work demonstrates the feasibility of controlling infrared light by nanostructures towards atomic level height. As a result, by engineering substrates vertically in atomic scale, like nano-etching or stacking layered materials, a variety of unique physical phenomena such as photonic crystal[22,23], Anderson localization[24,25], and functionalized optical devices[26,27] based on AGPs can be realized, finding potential applications including spectroscopy, sensing, and optoelectronics.

## Results

**s-SNOM imaging of the in-plane scattering of AGPs.** The schematic of our mid-infrared nanoimaging experiments on AGPs is shown in Fig. 1a. The heterostructure devices consist of a monolayer graphene flake encapsulated between two layers of hexagonal boron nitride (h-BN). The h-BN (2 nm)/graphene/h-BN (6 nm) heterostructure was assembled by the polymer-free van der Waals assembly technique[28]. Near-field imaging was realized by s-SNOM which is based on a metallic atomic force microscopy (AFM) tip[29–31]. The infrared laser with a wavelength

of 10.653 μm is focused on the tip, and the sharp tip apex provides wave vector matching between plasmons and incident photons, wherein the incident light is partly converted to plasmons[32,33]. The interference between the incident light and plasmon waves reflected back from graphene edges results in fringes with a period of one-half of the polariton wavelength.

Here, we placed the heterostructure on top of a 50 nm thick gold (Au) gate, where the bottom layer of h-BN acts as a spacer, and the GPs are supposed to hybridize with their mirror image to form AGPs. The propagating AGPs in our devices are visualized by infrared nano-imaging at room temperature, as illustrated in Fig. 1b, e at the back gate voltages −2 and −4 V, respectively. The most remarkable feature of the images is that the entire field of view (over 1.5 μm × 1 μm) is filled with plasmon interference fringes, which has never been reported at room temperature except for graphene under liquid-nitrogen temperatures[34]. However, it is worth mentioning that the physical origins are different. For the low-temperature graphene system, the fringes mainly depend on the reduced loss of plasmons, while obvious fringes in our scheme come from collective plasmon scattering by multiple scatterers. Moreover, these interference fringes are disorderly distributed and vary significantly at different gate voltages. It is well known that the interference between the in-plane scattering of propagating plasmons will result in fringes. To determine the scattering sources, the topography of the gold substrate is analyzed. AFM image of the uncovered gold surface reveals randomly distributed holes with depths around 6 nm (Supplementary Fig. 3a). Correspondingly, the morphology on the encapsulated sample region shows similar shaped but shallower holes, indicating a suspended device on the uneven substrate illustrated in Fig. 2a. Actually, the suspended graphene placed on a porous substrate has been studied for potential micro elctro-mechanical applications[35,36]. This kind of holes can be the scattering sources for the observed plasmon fringes, where a similar experiment but with a much larger depth of 180 nm has been reported[37]. Furthermore, to distinguish which these holes presented in the substrate here might stem from, i.e., non-uniform gold evaporation or possible chemical pollution during lift-off process, analysis techniques including scanning electron microscope (SEM) and energy-dispersive X-ray spectroscopy (EDS) were used to characterize the atomic composition of the sample (Supplementary Fig. 6). By comparing the characteristic peak intensities and atomic percentages of different elements, it can be deduced that the plasmon scattering is caused by the holes formed by gold. In other words, the metallic holes formed during nonuniform evaporation play the role of plasmon scattering sources. In further, the AFM topography of the samples and corresponding near-fields for the same area are compared in Supplementary Figs. 4 and 5, which confirms further the standing waves come from the plasmon scattering of these nanometer uneven of the substrate. Because the wavelength of the plasmon depends on the gate voltages, the difference of interference fringes between Fig. 1b and Fig. 1e can be understood as that different plasmon interactions appear between scatterers with the fixed spatial distribution.

To further confirm that the fringes originate from the interference of AGPs, the Fourier transformed near-fields are shown in Fig. 1c, f. The well fitted concentric circles in momentum space indicate that the chaotic fringes have wave vectors of identical magnitude but widely distributed directions. Besides, a roughly constant wavevector $k_x$ appears due to the background noise signal of the near-fields. From the radius of the circles, the obtained fringe periods Λ in Fig. 1b, e are 56 and 67 nm through $\Lambda = 2\pi/|\mathbf{k}|$, respectively. Simultaneously, the plasmons at the boundary of the h-BN encapsulated graphene with voltages of −2 and −4 V were also measured and the half

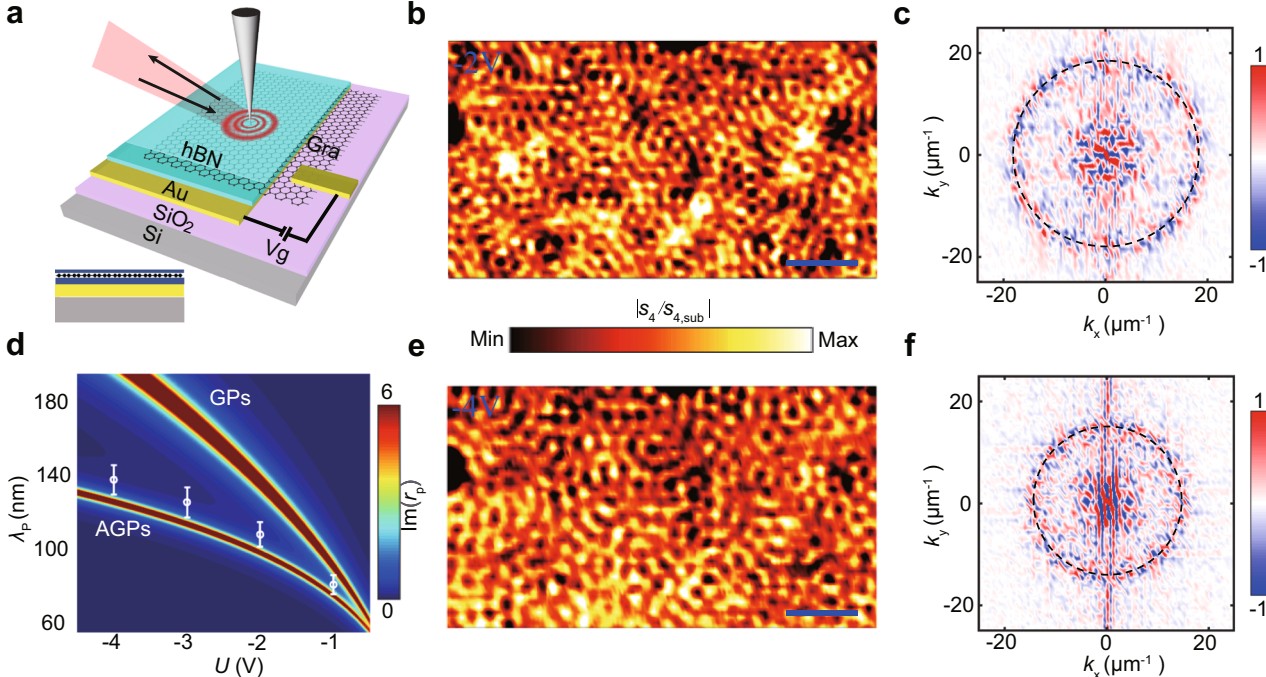

**Fig. 1 Structure and the near-field response of an acoustic graphene plasmon (AGP) device. a** Schematic of the AGP device, which is composed of h-BN (2 nm)/graphene/h-BN (6 nm) heterostructure placed on Au (50 nm)/SiO₂ (300 nm)/Si substrate. The inset shows the cross-section view of the AGP device. **b** Real space near-field image of the AGPs at −2 V, the color bar represents the normalized intensity. **c** The amplitude of the Fourier transformation of (**b**) in momentum space ($k_x, k_y$). **d** The comparison between the wavelength and back gate voltage-dependent dispersion of AGPs and that of graphene plasmons (GPs). The dispersion was obtained by caculating the imaginary of the reflection coefficient ($Im(r_p)$) with the transfer matrix method. The white circles represent the experimental results. The error bars show the width of the concentric circles in momentum space. **e**, **f** Near-field image of the AGPs at −4 V and the corresponding Fourier transform. The wavelength of excitation light is $\lambda = 10.653\,\mu m$. Scale bars, 300 nm.

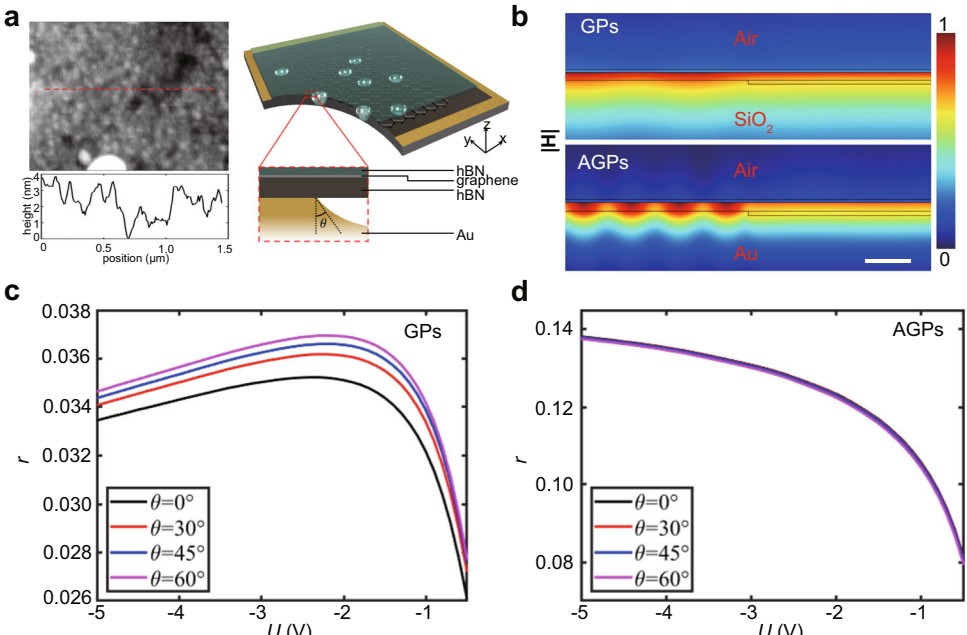

**Fig. 2 The advantage of in-plane reflectivity of mid-infrared acoustic graphene plasmons (AGPs) compared to graphene plasmons (GPs). a** Schematic diagram of the simplified calculation model. The left panels show the AFM topography of the h-BN/graphene/h-BN heterostructure placed on the gold surface, where a fluctuation in a depth of about 3 nm exists. Deduced from the topographic differences between the sample and bare gold substrate, the height of the reflection boundaries is set as 3 nm. The angle $\theta$ is the angle between the surface normal and the tangent of boundary on the left side of hole. **b** The magnetic field profile of |**H**| during reflection at $\theta = 0°$ for GPs and AGPs. Scale bar, 30 nm. **c**, **d** Show back gate voltage $U$ dependent reflectivity $r$ of GPs and AGPs for different steepness of steps, respectively.

wavelengths are approximately 54 and 68 nm, which are consistent with the above obtained periods. Besides, the plasmon response of h-BN/graphene/h-BN with different gate voltages is illustrated by its dispersion relation (Fig. 1d), where the plasmon wavelength $\lambda_p \propto E_F \propto \sqrt{|n|} \propto \sqrt{U}$, where $U$ is back gate voltage between Au and graphene. Compared to the traditional GPs, at the same gate voltages (the same $E_F$), the AGPs have a shorter wavelength with more localized field. The white circles on the color map are the experimental results, corresponding to voltages of −1, −2, −3, and −4 V, respectively, showing good agreement with theoretical ones. A small discrepancy may come from that the near-field images are dominated by the first two peaks close to the scattering sources, where the peak–peak distance is a bit larger than the plasmon wavelength[3]. This confirms further that the AGPs are the source of the interference fringes. It is worth noting that the dissipation of AGPs also depends on the gate voltage[14], however, it affects more on the dissipation time rather than wavevector, where the latter is obtained from experiments.

It is known that reflection is always used when a flat interface is illuminated by a wave. However, in our case, the AGPs incident on a hole-like structure instead of an interface, the scattering happens, and the near-field fringes come from the interference between the incident plasmon wave and backscattering wave. Compared to a complicated simulation for the 3D scattering model, an approximated 2D reflection model is much simpler and gives the same conclusions (Supplementary Fig. 10). Therefore, a theoretical 2D model is adopted to explore the physical reason why strong backscattering of AGPs happens by such shallow steps (approximately 3 nm) on the Au substrate. The scattering source of AGPs can be approximately imitated as holes in the metallic surface. Therefore, the reflectivity of traditional GPs and AGPs for a single scattering boundary was calculated through full-wave electromagnetic simulations by utilizing Comsol Multiphysics. Although the simplified model can not provide quantized efficiency for our sample, it can offer clues for understanding the physics behind it. The schematic diagram of the calculation model is shown in the right panel of Fig. 2a. In the simulation, the scattering boundary is described by a step. The height of the step is chosen as 3 nm. And $\theta$ represents the steepness of the step, where $\theta = 0°$ corresponds to the vertical boundary. For both cases, the h-BN (2 nm)/graphene/h-BN (6 nm) heterostructure is placed on top of the step. The GPs and AGPs with the same amplitude are input from the left port. Figure 2b shows the magnetic field $|\mathbf{H}|$ during reflection at $\theta = 0°$. One can clearly see that for the steps with the same depth, AGPs show an advantage over GPs, reflecting significantly while GPs hardly reflect. Moreover, the calculated reflectivity under different gate voltages is shown in Fig. 2c, d, which corresponds to GPs and AGPs, respectively. It can be known that, for a step with a height of 3 nm, the reflectivity of GPs is less than 3.8%, but it can reach about 13% for that of AGPs. This comes from much larger vertical field localization in the gap between metal and graphene for AGPs compared to GPs. At the same time, more energy is localized near the boundary in the AGPs case compared to the GPs case. It is worth mentioning that large reflection of traditional GPs can only happen at nanometer scale when the incontinuity of electronic conductivity of graphene is realized[38], however, this is not the requirement for AGPs. Besides, as revealed by the AFM morphological comparison between the sample (left panel of Fig. 2a) and the bare Au substrate, we know that the encapsulated region is partially suspended on the holes. Therefore, the reflectivity of the graphene surface with the similar nanometer steps should be considered. However, a minor effect on the reflection of the AGPs is raised when the graphene drops less than 20% (Supplementary Fig. 9).

**Electrical tuning on the scattering of isolate scatterers**. Instead of the collective plasmon interactions induced by randomly distributed scattering sources, the plasmon response of individual scattering source is important as well. To clearly understand the scattering behavior of individual scatterers, another sample showing spatially well-separated scatterers was studied. The sample was assembled in the same way as above, and the thicknesses of the top and bottom h-BN layers are 2 and 5.5 nm, respectively. Figure 3a shows the near-field image with sample region over a 2.6 μm × 2 μm area at a gate voltage of 2.9 V. The black dashed line on the right marks the boundary of the h-BN encapsulated graphene. One can find several isolated scatterers exist on the near-field image. And one of them is selected for an in-depth study as enclosed by the black dashed square in Fig. 3a. The height of the scatterer is approximately 3 nm with the width of 100 nm (Supplementary Fig. 11), which is consistent with previously proposed theoretical model (Fig. 2a). Plasmon near-field mappings on the selected region at gate voltages of 2.9 and 2.2 V are shown in Fig. 3b. The intensity of the fringes increases, and more fringes become visible at the gate voltage 2.9 V compared with that at 2.2 V. Moreover, the near-field profiles at different gate voltages across the same scatterer center (the scatterer 1) were extracted along the black dashed line in Fig. 3d. Obviously, three distinct peaks are observable when the voltage is 2.9 V, which are hard to be resolved at the voltage of 2.2 V. At the same time, the near-fields along the boundary of the encapsulated graphene are shown as the solid lines in Fig. 3d for comparison. We can find the consistent plasmon wavelengths for the substrate scatterers and graphene boundary, and both increase with the gate voltage.

It is noteworthy that obvious interference fringes appear in Fig. 3a, though the height of these scatterers changes less than 3 nm, which is about four orders of magnitude lower than the incident wavelength of light. Besides, the amplitude of the main fringe of the scatterers is not the same for different gate voltages. Therefore, we further studied the visibility of the main fringe between the scatterers and the boundary of the encapsulated graphene. Three obvious scatterers in the sample are selected and labeled by black arrows. Here, the visibility of the fringes is defined as $V = \Delta S_S / \Delta S_R$, where $\Delta S_S$ and $\Delta S_R$ are the magnitudes of the first peak in the near-field image relative to the near-field amplitude far away from it for the scatterer and boundary, respectively. $\Delta S_R$ keeps almost unchanged as the back voltage increases (Supplementary Fig. 12). However, the visibility gradually increases to 0.82 when the voltage changes from 1.6 to 2.9 V (Fig. 3c). These features can be understood for the following reasons. Firstly, the scattering of AGPs by holes is more divergent in directions than that for a boundary. This point is illustrated by the comparison of the angular resolved energy flow and the corresponding near-field distributions (Supplementary Figs. 13 and 14). As a result, the fringe visibility is always less than 1. Secondly, the larger visibility with higher voltages can be attributed to larger reflectivity of AGPs, which comes from the increase of Fermi energy of graphene by gate voltages (Fig. 2d). In further, for different scatterers, the visibility is not exactly the same under the same gate voltage, which may attribute to differences in the size and height of the scatterers.

## Discussion

Because of the unique properties that strong in-plane scattering of infrared light can happen even if the steps are close to atomic level, many applications can be envisioned. Firstly, the ultra-fine height sensors can be realized based on that the scattering amplitude of AGPs depends a lot on the height. Except for that, if

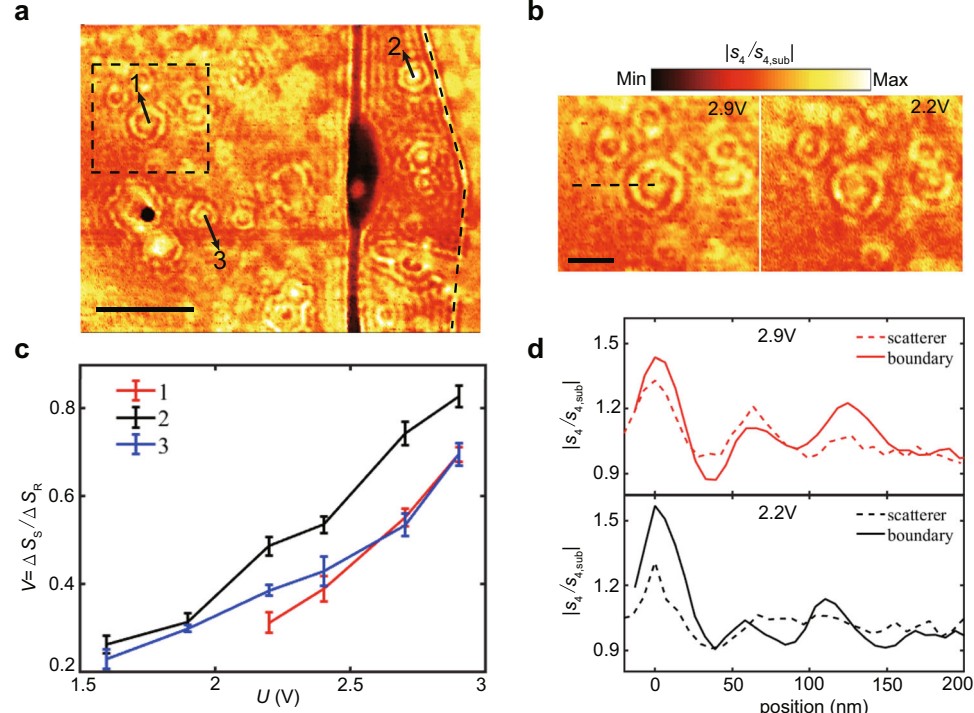

**Fig. 3 The near-field response of individual scatterers at the wavelength of $\lambda = 10.653\,\mu m$. a** The near-field image of the sample region over a $2.6\,\mu m \times 2\,\mu m$ area under the gate voltage of 2.9 V. The black dashed line and rectangle show the boundary and the selected scatterer area of the h-BN encapsulated single-layer graphene, respectively. Three obvious isolated scatterers are labeled as 1, 2, and 3 by arrows. Scale bar, 500 nm. **b** Near-field images of the acoustic graphene plasmons (AGPs) in the region of the black dashed square in (**a**) (the scatterer 1) at 2.9 and 2.2 V, respectively. The black dashed line represents the position where the detailed near-field closed to the scatterer is investigated in (**d**). Scale bar, 300 nm. **c** The fringe visibility ($V$) for the three different scatterers as a function of the gate voltages $U$. The errors bars show the fluctuation of $\Delta S_S/\Delta S_R$ in different azimuthal directions close to the scatterers. **d** Comparison between the normalized near-field signal $S_4/S_{4,sub}$ of the selected scatterer and the boundary of the graphene as a function of the voltages. The dashed and solid lines represent the scatterer and the boundary, respectively.

the hole region can be filled or replaced with few layer protein or chemical molecules with infrared absorption bands, AGPs always show better ultra-sensitive material sensing than GPs, as long as the the phonon frequency region of the interlayer hBN is avoided (Supplementary Fig. 15). Furthermore, our scheme may find applications in resolving intrinsically quantum mechanical effects with subnanometer resolution[39]. In addition, by controlling the parameters, strong scattering with multiple centers may enter the Anderson localization range, which could find applications in disorder physics.

In conclusion, real-space mid-infrared AGPs were observed in h-BN/graphene/h-BN heterostructure stacked on top of gold film with step height toward to atomic level. Specifically, at room temperature, extensive plasmon interference fringes were observed due to the scattering from randomly distributed nanometer deep steps of Au film, as not presented in traditional GPs, which can be attributed to that the reflectivity of AGPs is much larger than that of traditional GPs. Moreover, the individual scatterers show the obvious near-field response of AGPs, which can be efficiently tuned by the gate voltage. Our work is a preliminary optical study on 2D cavities with atomic height[21], although only single scattering event is explored, muti-scattering, coherent scatting, and even Anderson localization effect[40] can be expected in this system. Our work opens a venue for efficiently control acoustic plasmons with atomic level height nanostructures, which sheds light onto ultraconfined optical device design by atomic scale substrate engineering, finding potential applications including spectroscopy, sensing, and nanoscale lasers.

## Methods

**Samples and devices**. Graphene and h-BN were obtained by mechanical cleavage of bulk graphite and hexagonal boron nitride crystal and then transferred to SiO₂ (300 nm)/Si substrate. Then by employing a polymer-free van der Waals assembly technique[41]. Firstly, the top hBN on a SiO₂ (300 nm)/Si is brought in contact with a viscoelastic PDMS (polydimethysiloxane) stamp which has a polycarbonate (PC) film attached to it in a transfer stage arrangement. The top hBN is picked up by the PC film when it comes in a contact with PC film. Secondly, when the single graphene is brought in contact with the top hBN, it is picked up by top hBN due to vdW force between hBN and graphene greater than that between graphene and SiO₂ (300 nm)/Si substrate. Lastly, the top hBN/graphene assembly is aligned on top of the another hBN (down hBN: 6 nm) on SiO₂ (300 nm)/Si substrate and brought in contact with the flake. Then at an elevated temperature about 150°, the whole encapsulated graphene structure was placed on the Au (50 nm)/SiO₂ (300 nm)/Si substrate. The PC film is dissolved by putting the stack in a chloroform solution for 45 s at room temperature. Electrical contact to graphene and backgate were realized by pre-defined Au pads on Si/SiO₂ via UV lithography and electron-beam deposition.

**Plasmon near-field image**. Near-field imaging was achieved by the s-SNOM (Neaspec GmbH) based on a metal AFM tip, which is illuminated by infrared light from a CO₂ laser. Plasmon reflection at the graphene edges and scatterers produces plasmon interference, which is imaged by recording the light elastically scattered by the tip with a pseudo-heterodyne interferometer. In order to suppress background scattering from the tip shaft and sample, the tip is vibrated vertically with the frequency of around 270 kHz, with an oscillation amplitude of about 60 nm, and the fourth-order demodulated harmonics of the near-field amplitude was adopted.

**Dispersion calculation of AGPs**. The dispersion of AGPs was calculated using a classical electromagnetic transfer matrix method, with a thin-film stack of air/h-BN (2 nm)/graphene/h-BN (6 nm)/Au (50 nm)/SiO₂ (300 nm)/Si. Specifically, Fig. 1d is obtained by calculating the imaginary part of the reflection coefficient ($Im(r_p)$) of the evanescent wave. The conductivity of graphene was given by local random-phase approximation[42] and the h-BN permittivity parameters were shown in Supplementary Table 1.

## Data availability
The data that support the findings of this study are available from the corresponding author upon reasonable request.

## Code availability
All relevant calculation codes are available from the corresponding author upon reasonable request.

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

## Acknowledgements
This work was supported by Guangdong Major Project of Basic and Applied Basic Research (2020B0301030009), the Program for the National Key R&D Program of China (2017YFA0305100, 2017YFA0303800), the National Natural Science Foundation of China (91750204, 12074200, 11774185, 12004196, 61775106, 92050114), Changjiang Scholars and Innovative Research Team in University (IRT13_R29), the 111 Project (B07013), the Tianjin Natural Science Foundation (18JCQNJC02100), and Fundamental Research Funds for the Central Universities.

## Author contributions
W.C., W.L., and J.X. conceived the idea and supervised the project. N.Z. and W.L. performed the experiments. W. W. and M. R. participated in the sample characterization. Theoretical background and simulations were provided by L.W., J.F., and Z.Z. N.Z., W.L., and L.W. contributed equally to this work. All authors discussed the results and wrote the manuscript.

## Competing interests
The authors declare no competing interests.
