## [Peer Review File · Nature Communications]

Strong in-plane scattering of acoustic graphene plasmons by surface atomic stepsREVIEWER COMMENTS

Reviewer #1 (Remarks to the Author):

This manuscript presents the imaging of acoustic plasmons in encapsulated graphene. The experiments show plasmon standing waves in the interior of the graphene in addition to sample edges as studied previously. The authors attribute these standing waves to the scattering of surface roughness of the gold substrate and have conducted gating experiments. The high confinement of acoustic plasmon also contributes to the reflection of plasmons from the rough Au surface. Their experiments agree with the numerical simulation using COMSOL.

This work is interesting and useful. To meet the bar of the publications in Nature Communications, I listed my comments and suggestions below.

1. The origin of the plasmon standing waves needs further confirmation and clarification. First, the near field data was shown in FIG 1 while the AFM topography is in FIG 2, can these two be directly correlated or compared to show nanometer steps indeed scatter graphene plasmons?
2. Nanometer steps are described as scatters many times in the main text, but the modeling is based on the reflection of plasmons from these steps. Can this be clarified on whether it is the scattering or reflection as the origin of the observed results?
3. Based on the above comments and the stress of "atomic level steps" in the title, it may be necessary to show the dependence of step height in the formation of the plasmon standing wave. Will sub nanometer step (atomic step) indeed scatter or reflect plasmons?
4. There is a discrepancy between the experiments and simulation. The model in FIG 2 assumes the graphene to be flat. In their device, the graphene surface show similar nanometer steps like the gold substrate based on the topography in the supporting information.
5. On Page 2 of the main text, the authors stress the entire image filled with plasmon and this has never been observed before except for Reference 34 under low temperatures. Is it a fair comparison to previous studies since in this work, there are lots of scatters under the sample, so it is expected to see fringes everywhere? In Reference 34, many fringes are observed due to the lower loss under low temperatures.
6. On Page 6 of the main text, can the authors confirm and elaborate on "scattering of AGPs by holes is more divergent in directions than that for a boundary"?
7. The conclusion paragraph may elaborate more about the applications and perspective of this work.

Reviewer #2 (Remarks to the Author):

I have read the manuscript entitled: "Acoustic graphene plasmons advantage: in-plane strong scattering of infrared light by steps at the atomic level" by Zhang et. al.. In this work, they perform near-field imaging of acoustic graphene plasmons. In contrast to previous works, the development of SNOM in AGP systems is relatively new, with few other works on the topic. Moreover, their focus is on in-plane scattering from imperfections (in this case, shallow holes in the gold substrate). The fringes in Fourier space are consistent with the edge reflection (the latter of which is commonly observed) and also with the theoretical dispersion relation, leading to high-confidence that the signal is AGP in origin, They show a fairly robust room-temperature signal of in-plane scattering from defects, which they attribute (via 2D simulations) to high reflectivity coming from the concentration of EM energy closer to the gold. Finally, they study the visibility of a single fringe, attributing its relatively low visibility (compared to a boundary) to the non-directionality of scattering from a localized source.

Generally, the result seem fairly consistent (and the results are nice) -- much of what is observed is expected (though I have some questions below about fringe visibility, and a few others). They are expected on both theoretical grounds, as well as SNOM imaging in other systems, as well as several studies showing the high localization of AGPs (e.g., the seminal studies on the topic by the Koppens group). Thus, I feel that the added novelty provided by this work is insufficient to merit publication in Nature Communications, even though they consider AGPs. Besides this, I have some general

comments that the authors can consider when revising their work.

1. Fig. 1, the Fourier space maps: what is the origin of the lines of roughly constant k_x ?
2. Discussion around Fig. 2 and the reflectivity: it may be valuable to state at the given wavelength of interest how much more energy resides near the scatter in the AGP case compared to the GP case.
3. Fig. 3 discussion: the authors need to elaborate on the reasoning as to how the angular distribution of scattering of a point scatterer translates to smaller visibility. I can presume that the field has to decrease more rapidly in the point case, and that at the first maximum, the field is smaller. But this should be explained in more rigorous detail.
4. The manuscript needs to be proofread for proper English by an English-proficient editor. Many of the connectors between thoughts, as well as many words are mischosen. Much of this could be fixed at a later copy-editing stage, but it probably ought to be done upon resubmission as it does mar the reading.

Reviewer #3 (Remarks to the Author):

I have reviewed the manuscript titled as 'Acoustic graphene plasmons advantage: in-plane strong scattering of infrared light by steps at the atomic level'.

This work studied the acoustic plasmons in a graphene/dielectric/metal structure using scanning near-field optical microscope. They observe some fringe patterns that are attributed to the acoustic plasmons. However, the presented result is far from sufficient to support their claim.

For instance, (i) what is the acoustic plasmon dispersion looks like? (ii) why the acoustic plasmons that is normally emerging in the THz regime can show up at the mid-infrared frequencies? (iii) how does its dissipation varies as a function of the external stimulus? (iv) what is the pros and cons of acoustic plasmons compared with normal plasmons at this frequency etc?

None of these important questions was addressed or even touched yet. After reading the MS, the reviewer does not feel learn much new from it. They spend a length to explain that these fringes patterned are initiated by the non-uniform Au protrusions. But it is unclear what is new physics or potential implications of these studies.

It is also noticed that acoustic phonon studies in sandwiched graphene using the same technique and identical device structures have been reported in several other journals including Nature Comm itself. Thus it is hardly to be convinced that the current work is suitable for the next steps.

Response to Reviewer #1

“This manuscript presents the imaging of acoustic plasmons in encapsulated graphene. The experiments show plasmon standing waves in the interior of the graphene in addition to sample edges as studied previously. The authors attribute these standing waves to the scattering of surface roughness of the gold substrate and have conducted gating experiments. The high confinement of acoustic plasmon also contributes to the reflection of plasmons from the rough Au surface. Their experiments agree with the numerical simulation using COMSOL.

This work is interesting and useful. To meet the bar of the publications in Nature Communications, I listed my comments and suggestions below.”

---Reply. We thank the referee for the objective evaluation of our work.

--- Referee 1, point 1. “The origin of the plasmon standing waves needs further confirmation and clarification. First, the near field data was shown in FIG 1 while the AFM topography is in FIG 2, can these two be directly correlated or compared to show nanometer steps indeed scatter graphene plasmons?”

--- Reply. We have checked the experimental data again and show that the standing waves come from the plasmon scattering of nanometer steps. To confirm this point, the AFM topography of sample 1 and corresponding near-fields for the same area are given in Fig.r1. From the wide field of view, the correlation between steps and near-field signal is not obvious. As a result, two independent regions labelled as I and II were chosen. The line distribution of the AFM and near-field data in these two regions are given in Fig.r1(c)-(d) and Fig.r1(e)-(f), respectively. One can deduce the near-fields signal appears due to unflatness of the surface, because it exists in both directions away from the steps. However, the signal is not consistent with a typical reflection of plasmons by a boundary, which can be attributed to the collective multiple scattering due to the large density of the scatterers for the sample 1. As a result, the sample 2 with a low density of scatterers is examined. Two independent scatterers were selected and shown in Fig.r2. Figure r2 (a) and (b) are the AFM and corresponding near-field of the independent scatterers I and II. From the AFM morphology we can clearly see the position of the scatterer, where a hole-like structure is formed. Fig.r2(c)-(d) and (e)-(f) show the line profile of the AFM, and the near-field profile shown by the white dotted line in Fig.r2(a)-(b). It is obvious from the field distributions that the fringes are caused by the scattering of scatterers, where near-fields exist outside the hole. Except for that, inside the hole region, the plasmon resonance also exists, which contributes additional momentum components in the dispersion plots in Fig. 1 (c) and (f) in the main text.

(a)

(b)

Fig.r1: The AFM topography and the corresponding near-field image of the Fig. 1 in the main text. (a) The AFM topography. (b) The near-field image of (a). (c)-(d) and (e)-(f) is the line profile of the AFM and the near-field profile of I and II.

(a)

(b)

Fig.r2: The AFM topography and the corresponding near-field image of two independent scatterers. (a), (b) The AFM topography and the near-field image of the independent scatterers of I and II. (c)-(d) and (e)-(f) is the line profile of the AFM and the near-field profile shown by the white dotted line in I and II.

We thank the review for the comment. On Page 3 of the main text, we have added: “The AFM topography of the samples and corresponding near-fields for the same area are

given in supplementary (see Fig.~S4 and Fig.~S5 in supplementary), which confirm further that the standing waves come from the plasmon scattering of these nanometer uneven of the substrate”.

--- **Referee 1, point 2.** “Nanometer steps are described as scatters many times in the main text, but the modeling is based on the reflection of plasmons from these steps. Can this be clarified on whether it is the scattering or reflection as the origin of the observed results?”

--- **Reply.** Sorry for the confused description. The observed results actually come from the back scattering of acoustic plasmons by the hole-like structures. It is well known that reflection is used when a flat interface is illuminated by a wave. However, in our case, the acoustic graphene plasmons incident on a nonhole-like defect instead of an interface, the scattering happens, and the near-field fringes come from the interference between the incident plasmon wave and backscattering wave. Compare to complicate 3D simulation for the scattering event, an approximated 2D reflection model was adopted in the work for simplicity. And both models give the same conclusions. To prove this point, the 3D scattering model is performed here, scattering length instead of reflection coefficient is obtained. The model is shown in Fig.r3(a), where the incident plasmons is scattered by a circular nanohole with the height of 3 nm and the diameter of 100 nm. The integral of the scattering energy flow is calculated. Here, the scattering energy flow is defined as $S=0.5*(E-E_0)\times\text{conj}(H-H_0)$, E_0 and H_0 is the electric and magnetic near-fields of graphene plasmons without small holes. The integration area is chosen as a cylindrical area with a radius of $0.1\mu\text{m}$ at the bottom of the cylinder and a height of $0.26\mu\text{m}$ (Fig.r3(b)). Taking into account the loss of graphene plasmons, the integrated energy flow is different in different integration ranges, therefore, the loss of graphene is ignored in the calculation. The calculated energy flow integral is defined as P . At the same time, the energy flow without the hole is integrated on the cylindrical cross section ($0.2\mu\text{m}\times 0.26\mu\text{m}$) to get the incident energy flow P_0 , and the final scattering length is defined as $L=P*(0.2\mu\text{m})/P_0$. The field distribution of the plasmons when scattering by a circular nanohole is shown as Fig.r3(c). And the scattering length is shown in Fig.r3(d). The normalized scattering (divided by the diameter of the hole) length larger than 1 can be obtained for acoustic plasmons, which is difficult for traditional graphene plasmons. The valley corresponds to the position of the hBN phonon. At the excitation wavelength of $10.653\mu\text{m}$ (shown as the black dotted line), the scattering length of the AGPs is about 6.5 times than that of the GPs. From that it can be known the scattering ability of AGPs is much stronger than that of GPs. The result is consistent with the simplified reflection model.

(a)

(b)

Fig.r3: The scattering length of AGPs and GPs by a nanohole with a height of 3 nm. (a) Schematic diagram of scattering of the plasmons for an independent scatterer. (b) The model of calculating the energy flow integral. (c) The field distribution of the plasmons E_z when passing through a circular nanohole. And the black dotted circle represents the point scatterer. (d) The scattering length of AGPs and GPs. The black dotted line corresponds to 10.653 μm .

On Page 3 of the main text, we have added: “It is known that reflection is always used when a flat interface is illuminated by a wave. However, in our case, the AGPs incident on a hole-like structure instead of an interface, the scattering happens, and the near-field fringes come from the interference between the incident plasmon wave and backscattering wave. Compared to a complicated simulation for the 3D scattering model, an approximated 2D reflection model is much simpler and gives the same conclusions (see Fig.~S10 in the supplementary). Therefore, a simplified theoretical 2D model is adopted to explore the physical reason why strong scattering of AGPs happens by such shallow steps (approximately 3 nm) on the Au substrate.....”.

--- **Referee 1, Point 3.** “Based on the above comments and the stress of “atomic level steps” in the title, it may be necessary to show the dependence of step height in the formation of the plasmon standing wave. Will sub nanometer step (atomic step) indeed

scatter or reflect plasmons?”

--- **Reply.** Thank you for your comments. It is difficult to accurately control the step height in subnanometer scale in our experimental scheme, however, the numerical simulation shows that acoustic graphene plasmons and graphene plasmons can be scattered by nanoholes with the height at atomic scale, but stronger scattering exists for acoustic plasmons. Fig.r4 shows the relationship between the length of the scattering cross section ($L=P*0.2\mu\text{m}/P_0$) and the depth of the hole (d). And the simulation model is the same as Fig. r3. Here, the Fermi energy is 0.3eV and the excitation wavelength is $10.653\mu\text{m}$. From the simulation results, it can be known that the scattering length decreases as the decrease of depth. Specifically, when the step is 0.3 nm, the length of the scattering cross section is 3.1 nm for the AGPs, but only 0.29 nm for the GPs, demonstrating much stronger scattering of AGPs than GPs. On the experimental side, it may be possible to verify these effects in two-dimensional empty spaces, such as the sample configuration in Nature 538, 222 (2016).

Fig.r4: The relationship between the length of the scattering cross section and the depth of the scatterer of AGPs and GPs.

--- **Referee 1, point 4.** “There is a discrepancy between the experiments and simulation. The model in FIG 2 assumes the graphene to be flat. In their device, the graphene surface show similar nanometer steps like the gold substrate based on the topography in the supporting information.”

--- **Reply.** We agree with the referee that the suspended graphene is not flat due to the gravity, and the graphene surface has the similar nanometer steps like the gold substrate. To consider the effect on the acoustic plasmon reflection, the reflectivity of the graphene surface with the similar nanometer steps like the gold substrate (Fig.r5(a)) have been simulated and shown in Fig.r5(b).

(a)

(b)

Fig.r5: The reflectivity of the graphene surface with the nanometer steps. (a) The schematic diagram of the simulation model and h_1 represent the height of the graphene surface recessed downward; (b) the reflectivity of the graphene surface with different h_1 . Here, $h_1=(5\%, 10\%, 15\% \text{ and } 20\%) h$, h is the height of the shallow steps.

By comparing the reflectance of the flat graphene and that of the graphene surface with a small drop, it can be known that, in such a system, whether the graphene is completely flat has a minor effect on the reflectivity of AGPs when the graphene drops less than 20%. This minor effect can be understood as following. The reflectivity depends on the impedance mismatch between acoustic plasmons and graphene plasmons in the suspended region. Small variation of the suspended region leads to minor effect on the graphene plasmons due to its strong field confinement. Therefore, assuming the graphene to be flat is a good approximation in our model.

On Page 4 of the main text, we have added: “Besides, as revealed by the AFM morphological comparison between the sample (left panel of Fig.~\ref{fig2}(a)) and the bare Au substrate, we know that the encapsulated region is partially suspended on the holes. Therefore, the reflectivity of the graphene surface with the similar nanometer steps should be considered. However, a minor effect on the reflection of the AGPs is raised when the graphene drops less than 20% (see Fig.~S9 in supplementary).”.

--- Referee 1, Point 5. “On Page 2 of the main text, the authors stress the entire image filled with plasmon and this has never been observed before except for Reference 34 under low temperatures. Is it a fair comparison to previous studies since in this work, there are lots of scatters under the sample, so it is expected to see fringes everywhere? In Reference 34, many fringes are observed due to the lower loss under low temperatures.”

--- Reply. Thank you for your comments. We admit that these contents are very loosely expressed. To be precise, these two works have similar experimental phenomena - lots of interference fringes, but with different physical origins. In Ref. 34, For low-temperature systems, the fringes mainly depend on the reduced loss of plasmon, where the inherent plasmon propagation length may exceed 10 microns or 50 plasmon wavelengths, thus setting a record for highly restricted and tunable polariton modes. For our system, obvious fringes are formed after collective scattering due to multiple scatter centers.

On Page 2 of the main text, we have added: “Which have never been observed at room temperature except for graphene under liquid-nitrogen temperatures, However, it is worth mentioning that the physical origins are different. For the low-temperature graphene system, the fringes mainly depend on the reduced loss of plasmons, while obvious fringes in our scheme come from collective plasmon scattering by multiple scatterers. Moreover, these interference fringes are disorderly distributed.....”

--- **Referee 1, point 6.** “On Page 6 of the main text, can the authors confirm and elaborate on “scattering of AGPs by holes is more divergent in directions than that for a boundary?”

--- **Reply.** Thank you for your comments. To illustrate this problem, under the model in Fig.r3, the angular distribution of the energy flow as passing the scatterer as shown in Fig.r6 (c) and (d). Here, the scattering energy flow is defined as the same as above: $S=0.5*(E-E_0) \times \text{conj}(H-H_0)$, E_0 and H_0 is the electric and magnetic fields of acoustic graphene plasmons without small holes. Fig.r6(a) and Fig.r6(b) show the schematic diagrams of calculating the scattered energy flow of the hole and a boundary, respectively. The intensity P of the scattered energy flow at a distance of $0.5\mu\text{m}$ in the horizontal direction away from the center is calculated. Taking the forward direction as 180° ($-x$ direction). The calculation results show that for the hole, the scattered energy flow is mainly in the range of polar angle from $-\pi/4$ to $\pi/4$; while for the plane, it mainly diverges in the forward and backward directions and only within an area with a small angle. This shows that the hole scattering is indeed more divergent than plane scattering.

(a)

(b)

(c)

(d)

Fig.r6: The angle distribution diagram of the energy flow when passing through a single hole and the flat boundary. (a) and (b) is the schematic diagrams of calculating the scattered energy flow of a single scatterer (hole) and the graphene boundary, respectively. (c) The angular distribution of energy flow for a single scatterer; (d) The angular distribution of the energy for a boundary of graphene.

On Page 7 of the main text, we have added: “Firstly, the scattering of AGPs by holes is more divergent in directions than that for a boundary. This point is illustrated by the comparison of the angular resolved energy flow and the corresponding near-field distributions (Fig.~S13 and Fig.~S14 in supplementary). As a result, the fringe visibility is always less than 1.”.

--- **Referee 1, point 7.** “The conclusion paragraph may elaborate more about the applications and perspective of this work.”

--- **Reply.** Thank you for your comments. Compared with traditional graphene plasmons, AGPs can be used as ultrasensitive sensors due to extremely confined electric fields. For our work, one of the most obvious applications is the atomic scale height sensor, in which the height of the steps can be detected by the near-field signals. Besides, if the scattering hole is replaced with protein or chemical molecules, the ultrathin layer can be effectively recognized by AGPs. The AGPs have the potential for quantitative protein detection and chemical-specific molecular identification. To show this potential application, the transmittance of acoustic plasmons by an atomic height protein layer is simulated. In the calculations, the analytic model is used to retrieve the protein permittivity from experimental results by adjusting a Lorentzian permittivity:

$$\varepsilon_p(\omega) = n_\infty^2 + \frac{s_1^2}{\omega_1^2 - \omega^2 - i\omega\gamma_1} + \frac{s_2^2}{\omega_2^2 - \omega^2 - i\omega\gamma_2}, \quad n_\infty^2 = 2.08,$$

$$\omega_1 = 1668\text{cm}^{-1}, \omega_2 = 1532\text{cm}^{-1}, \gamma_1 = 78.1\text{cm}^{-1}, \gamma_2 = 101\text{cm}^{-1}, s_1 = 213\text{cm}^{-1}, s_2 = 200\text{cm}^{-1}$$

Fig.r7(a) and (b) shows that when AGPs and GPs pass through a nano-scale (the height of protein is 0.5 nm, 1.0 nm and 1.5 nm) protein molecule (pyridine), AGPs will form very strong transmittance peaks at the two natural frequencies of the protein molecule, but the GPs will only form absorption peaks at ω_1 under the same circumstances. The transmittance of AGPs is significantly higher than that of GPs. It shows that AGPs have great advantages in biosensing for nano-scale protein compared to GPs.

Fig.r7: The absorption spectra of the AGPs and GPs when pass through protein molecules of different heights at 0.5, 1.0 and 1.5 nm.

On Page 7 of the conclusion part, we added: “Because of the unique properties that strong in-plane scattering of infrared light can happen even if the steps are close to atomic level, many applications can be envisioned. Firstly, the ultra-fine height sensors can be realized based on that the scattering amplitude of AGPs depends a lot on the height. Except for that, if the hole region can be filled or replaced with few layer protein or chemical molecules with infrared absorption bands, AGPs can be exploited in ultra-sensitive material sensing (Fig. S15 in supplementary). In addition, by controlling the parameters, strong scattering with multiple centers may enter the Anderson localization range, which could find applications in disorder physics.”

Response to Reviewer #2

“I have read the manuscript entitled: "Acoustic graphene plasmons advantage: in-plane strong scattering of infrared light by steps at the atomic level" by Zhang et. al.. In this work, they perform near-field imaging of acoustic graphene plasmons. In contrast to previous works, the development of SNOM in AGP systems is relatively new, with few other works on the topic. Moreover, their focus is on in-plane scattering from imperfections (in this case, shallow holes in the gold substrate). The fringes in Fourier space are consistent with the edge reflection (the latter of which is commonly observed) and also with the theoretical dispersion relation, leading to high-confidence that the signal is AGP in origin, They show a fairly robust room-temperature signal of in-plane scattering from defects, which they attribute (via 2D simulations) to high reflectivity coming from the concentration of EM energy closer to the gold. Finally, they study the visibility of a single fringe, attributing its relatively low visibility (compared to a boundary) to the non-directionality of scattering from a localized source. the visibility of a single fringe, attributing its relatively low visibility (compared to a boundary) to the non-directionality of scattering from a localized source.

Generally, the result seem fairly consistent (and the results are nice) -- much of what is observed is expected (though I have some questions below about fringe visibility, and a few others).”

---- **Reply.** We thank the referee for confirming the correctness of our work.

“ They are expected on both theoretical grounds, as well as SNOM imaging in other systems, as well as several studies showing the high localization of AGPs (e.g., the seminal studies on the topic by the Koppens group). Thus, I feel that the added novelty provided by this work is insufficient to merit publication in Nature Communications, even though they consider AGPs. Besides this, I have some general comments that the authors can consider when revising their work.

--Reply. Thank you for your comments. Due to the unique properties of AGPs, many works have been explored in these field on both theoretic and experimental sides. However, the experiment work on AGPs is still few to the best of our knowledge. In the following, the related experimental works are listed in the following table.

	Work	Experimental method	novelty
Ref.1	Nat. Nanotech. 12, 31 (2017) (Hillenbrand group)	THz Near-field photocurrent nanoscopy	real-space imaging of acoustic THz plasmons in a graphene photodetector with split-gate architecture by nanoscale-resolved THz photocurrent near-field microscopy
Ref. 2	Nat. Nanotech. 14, 313(2019) (Sang-Hyun Oh group)	Mid-infrared FTIR	Realize a graphene acoustic plasmon resonator with nearly perfect absorption (94%) of incident mid-infrared light
Ref.3	Science 357, 189 (2017) (Koppens group)	THz near-field photocurrent nanoscopy	probe the nonlocal response of the graphene electron liquid by acoustic plasmons
Ref.4	Science 368, 1219 (2020) (Koppens group)	Mid-infrared FTIR	Single acoustic plasmon cavity with ultracompressed mode volumes
Ref.5	Science 360, 291 (2018) (Koppens group)	Mid-infrared FTIR	plasmon confinement down to the ultimate limit of the length scale of one atom
Ref.6	Nat. Commun. 12, 938 (2021) (Min Seok Jang group)	Mid-infrared near-field optical microscopy	acoustic plasmons in large-area graphene grown by chemical vapor deposition
	Our work	Mid-infrared near-	Strong scattering of

		field microscopy	optical	acoustic plasmon by atomic steps
--	--	---------------------	---------	-------------------------------------

As we can see, Koppens' group indeed have done many excellent works in the field of acoustic graphene plasmons (Ref.3-5), they have shown isolated acoustic plasmon cavities with extremely field confinement and used acoustic graphene plasmon to probe the extremely field localization and quantum effect of electrons. However, far-field optical spectrum or near-field photoncurrent nanoscopy is used, which are obviously different from our work.

Moreover, Ref1 from Hillenbrand's group have demonstrated that near-field photocurrent nanoscopy can be used for polariton exploration at THz range. Ref2 provides a two-step coupling mechanism to realize over 90% absorption of infrared light and proves that acoustic plasmons can be used for ultrathin protein analysis. Ref.6, which is published after our submission, the only work which takes the same experimental tools as us, but the major novelty of the paper is that large area CVD graphene shows good acoustic plasmon response as well compared with mechanical exfoliated graphene.

Though the study of AGPs have been mentioned, we would like to stress that unlike the impression the Referee may have had, our manuscript is contributing to the in-plane scattering of AGPs, which has never been mentioned, and it is essential for the manipulation and utilization of ultraconfined optical field down to atomic level. As a result, we believe our work should be competitive for Nat. Commun., and hope he/she will consequently reconsider the recommendation.

In the revised version, we have stressed more on our novelty compared with other work in paragraph 2 in the Introduction part as "Yet, though lots of work on AGPs have been reported, the experimental progress is still few. Specifically, both near-field photocurrent nonoscopy [15,17] and far-field infrared spectrum [12, 16, 18] are used to explore the properties of AGPs in either THz or infrared ranges, however, the direct detection of AGPs through infrared near-field optical scattering [21] is still difficult, especial on the in-plane scattering properties of AGPs.. And this is essential for the manipulation and utilization of ultraconfined optical field down to atomic level."

--- **Referee 2, point 1.** "Fig. 1, the Fourier space maps: what is the origin of the lines of roughly constant k_x ?"

--- **Reply.** Thank you for your comments. After careful examination, we find the lines of roughly constant k_x originates from the noise background signal of the near-field. From the Fig. 1 in the main text, the brightness in the near-field image is not uniform. Obviously, the intensity in the lower left corner is stronger than that in the upper right corner. Therefore, we deduced that the vertical line in the Fourier transform comes from the ununiform background field strength distributions.

To prove this point, a mathematical averaging process is performed on Fig. 1 of the main text. Firstly, the average intensity distribution along the x-direction of Fig. 1 is shown as Fig.r8(a), and there is an obvious change in the intensity distribution along the y-direction. Use a straight line to do a simple linear fit to the trend of the scattered points, as shown by the straight line in Fig.r8(b).

Fig.r8: The background signal with uneven intensity. (a) The uneven intensity of the background signal. (b) The extracted background signal data (blue circle) and the corresponding linear fitting (red line).

Then we use Fig. 1 subtract the straight line in Fig.r8(b) to obtain near-field image without the noise background. Then the corresponding Fourier transform is shown in Fig.r9. Comparing the Fourier transform diagrams before and after processing, it can be seen that the vertical line in the middle has become much weaker. The reason why it is not eliminated is because the noise background is only a rough linear fit.

Fig.r9: The Fourier transform of the near-field image without the noise background.

To further illustrate the relationship between the vertical line and the intensity, a ratio of the two Fourier transformation spectra in Fig. 1 and Fig.r9 is made. It can be found that this vertical line obviously comes from the ununiform intensity distribution after

the transformation. Although the intensity distribution of the original image is not uniform, the original data collected as an experiment does not affect the analysis of the results.

Fig.r9: The ratio of the Fourier transform spectrum before and after noise background removal.

In the revised version, we have added in Paragraph 3 of section 2, “The well fitted concentric circle in momentum space indicates that the chaotic fringes have wave vectors of identical magnitude but widely distributed directions. Besides, the roughly constant k_x originates from the noise background signal of the near-fields. From the radius of the circles, the obtained fringe periods.....”

--- **Referee 2, point 2.** “Discussion around Fig. 2 and the reflectivity: it may be valuable to state at the given wavelength of interest how much more energy resides near the scatter in the AGP case compared to the GP case.”

--- Reply. Thank you for your comments. We have integrated the energy flow of GPs and AGPs when pass through the step, respectively. Fig.r10(a) is the schematic diagram for calculating the total inflow (no step) and (b) is the model for calculating the outflow energy flow. For GPs the inflow is 0.24206W; the outflow is -0.12929W; for AGPs the inflow is 0.23101W; the outflow is -0.12729W. GPs:0.12929/0.24206=53.4%; AGPs:0.12729/0.23101=55.1%. It can be shown that more energy is localized near the scatterer in the AGP case compared to the GP case. From another perspective, this result is consisted with the calculations that stronger scattering happens for AGPs and GPs, (Point2, referee1), in which stronger polarization or higher charge density distributions appears for AGPs.

Fig. r10. The schematic diagram of the simulation model of the inflow and outflow energy flow, respectively. (a) The simulation model of inflow energy flow. The red dotted line is the integral line. (b) The simulation model of outflow energy flow. The red dotted box is the integration area.

In the revised version, we have added in Paragraph 5, “This comes from much larger vertical field localization in the gap between metal and graphene for AGPs compared to GPs, providing a great advantage for AGPs. At the same time, more energy is localized near the scatterer in the AGP case compared to the GP case. It is worth mentioning that large reflection of traditional GPs.....”

--- **Referee 2, point 3.** “Fig. 3 discussion: the authors need to elaborate on the reasoning as to how the angular distribution of scattering of a point scatterer translates to smaller visibility. I can presume that the field has to decrease more rapidly in the point case, and that at the first maximum, the field is smaller. But this should be explained in more rigorous detail.”

--- **Reply.** First, the angular distribution is more divergent for plasmon scattering by a point source than a flat boundary, which have been shown in the reply to **Point 6 from the referee 1**. Moreover, we agree with the referee that the smaller visibility comes from the more rapidly decay of fields than a boundary. To illustrate this claim in a more rigorous way, using the model shown by the schematic in Fig.r6, where propagating plasmons are scattered by a hole and a graphene/air boundary, the electric field $|E_z|$ distribution at $z=0.4$ nm and $y=0$ nm is simulated. The results are shown in Fig.r11. Considering that the s-SNOM signal is formed by the coherent superposition of the incident wave and the reflected wave, we plot the electric field $|E_z|$ at position from $(-0.5\mu\text{m}, 0, 0.4e^{-3}\mu\text{m})$ to $(0\mu\text{m}, 0, 0.4e^{-3}\mu\text{m})$ in the backscattering direction of -180° . We can clearly see that the electric field decays more rapidly for the point source than the boundaries, which leads a smaller visibility for the point source.

Fig.r11: The field distribution along the center position for a point scatterer and a boundary, respectively.

On Page 7 of the main text, we have added: “the scattering of AGPs by holes is more divergent in directions than that for a boundary. This point is illustrated by the comparison of the angular resolved energy flow and the corresponding near-field distributions (Fig.~S13 and Fig.~S14 in supplementary). As a result, the fringe visibility is always less than 1.”

--- **Referee 2, point 4.** “The manuscript needs to be proofread for proper English by an English-proficient editor. Many of the connectors between thoughts, as well as many words are mischosen. Much of this could be fixed at a later copy-editing stage, but it probably ought to be done upon resubmission as it does mar the reading.”

--- **Reply.** We have done our best to modify the language and hope it is easier to read.

Response to Reviewer #3

I have reviewed the manuscript titled as ‘Acoustic graphene plasmons advantage: in-plane strong scattering of infrared light by steps at the atomic level’. This work studied the acoustic plasmons in a graphene/dielectric/metal structure using scanning near-field optical microscope. They observe some fringe patterns that are attributed to the acoustic plasmons. However, the presented result is far from sufficient to support their claim.

--- **Reply.** We are sorry for inaccurate description of our results in the paper. Following the comments, we have tried our best to make the claim clearer. Specifically, the dispersion relation of AGPs modes and traditional GPs at the excitation wavelength of 4.5-200 μm is given as below (Fig.r12). At the same time, the wavelength and back gate

voltage dependent dispersion of AGPs and GPs modes at the excitation wavelength of 10.653 μm also been simulated shown as below (Fig.r14), the wavelength of the plasmons satisfies the AGPs dispersion relation. So, these fringe patterns come from the interference of the acoustic graphene plasmons.

--- **Referee 3, point 1** “For instance, (i) what is the acoustic plasmon dispersion looks like?”

--- Reply. Thank you for your comment. In fact, the dispersion of acoustic graphene plasmon is determined by the following three parameters, the incident wave frequency, the wavevector of the plasmons, and the conductivity of graphene. We agree with the referee that we didn't provide the acoustic plasmon dispersion in a traditional way, where the varied incident wavelengths are needed and have been reported in previous work, such as in Nat. Nanotech. 12, 31 (2017), and Nat. Commun. 12, 938 (2021). However, the dispersion of AGPs is shown in Fig. 1(d) in an indirect way, where the plasmon wavevector varies depending on the Fermi energy related graphene conductivity at the same incident wave frequency. From which, we can understand the near-field fringes come from the interface of acoustic graphene plasmons.

--- **Referee 3, point 2** “(ii) why the acoustic plasmons that is normally emerging in the THz regime can show up at the mid-infrared frequencies?”

--- **Reply.** Thank you for your comment. When graphene is placed close to a metal, the coupling of graphene plasmon and its image leads to the hybrid plasmon branches. The low energy branch is labelled as acoustic mode, while the high energy branch is the optical branch. The acoustic graphene plasmons can show up not only in the THz range, but also at the mid-infrared frequencies, Fig.r12 shows the dispersion diagram including THz and mid-infrared frequencies, here $\lambda = 4.5 - 200 \mu\text{m}$, and $\omega = 2\pi c / \lambda$ is the vertical coordinates. we can clearly see that the mode also exists in mid-infrared range. The white dotted line represents the excitation wavelength of our experiments (10.653 μm). And the other two blue dotted lines corresponds to the frequencies of the phonons of hBN.

Fig.r12: The dispersion relation of AGPs modes at the excitation wavelength of 4.5-200 μm .

In addition, there already have been a lot of work reported at the mid-infrared frequencies, which have been listed as the following. The list of the acoustic plasmons show up at the mid-infrared frequencies reported before:

1. [(Epstein, Itai, et al., Science 368, 6496 (2020))]
2. [(Iranzo, David Alcaraz, et al., Science 360, 6386 (2018))]
3. [(Echarri, A. Rodríguez, et al., Optica 5, 6 (2019))]
4. [(Lundeberg, Mark B., et al., Science 357, 6347 (2017))]
5. [(Chen S., et al., ACS Photonics 12, 4 (2017))]
6. [(Lee, In-Ho, et al., Nat Nanotechnol 4, 14 (2019))]

In conclusion, the frequency range of AGPs is different from the referee's impression, which can show in infrared range.

--- **Referee 3, point 3.** (iii) how does its dissipation varies as a function of the external stimulus?"

--- Reply. Thank you for your comment. In our experiments, the external stimulus comes from the gate voltages on the graphene. As the gate voltage changes, the acoustic

graphene plasmons amplitude decay time satisfies the form: $\tau = \frac{\mu E_F}{eV_F^2}$, $E_F = \eta V_F \sqrt{\pi |n|}$,

$n = \frac{\epsilon_0 \epsilon_{BN} (V - V_0)}{dq_e}$, here, μ is the mobility of the graphene, $\epsilon_0 = 8.854 \times 10^{-12}$ is the

vacuum dielectric constant, $\epsilon_{BN} = 3.56$ is the electrostatic permittivity, $V_0 = 0.2V$ is

the inherent doping of graphene, d is the thickness of the down hBN, q_e is the

charge of the electron. By using the method of numerical simulation, the relaxation time of the plasmons shown as the following under: $\mu = 10000\text{cm}^2/\text{Vs}$ and $\mu = 12000\text{cm}^2/\text{Vs}$, respectively. It can be known that the acoustic graphene plasmons amplitude decay time increases as the gate voltages increases. This is consistent with the other work reported before [(Alonso-Gonzalez, P., et al., Nat Nanotechnol 1, 12 (2017))].

Fig.r13: The relaxation time (τ) of the plasmons with the change of gate voltages. (a) $\mu = 10000\text{cm}^2/\text{Vs}$; (b) $\mu = 12000\text{cm}^2/\text{Vs}$, and the other parameters keep consistent.

In the revised version, we have added in Paragraph 3, “This confirm further that the AGPs are the source of the interference fringes. It is worth noting that the dissipation of acoustic plasmons also depend on the gate voltages [15], however, it affects more on the dissipation time rather than wavevector, where the latter is obtained from experiments.”

--- Referee 3, point 4. “(iv) what is the pros and cons of acoustic plasmons compared with normal plasmons at this frequency etc?”

--- Reply. Thank you for your comment. In general, the comparison between normal graphene plasmons and acoustic graphene plasmons have been reported in many works. Due to larger momentum of AGPs compared with GPs, extremely confined electromagnetic fields can be obtained. AGPs have found for lots of applications, such as the nonlocal response of the graphene electron liquid (Science, 357, 189 (2021)), confinement down to the ultimate limit of the length scale of one atom (Science, 360, 291 (2018)), and ultrathin protein layer sensor (Nat. Nanotech,14, 313 (2019)).

Specially for the AGPs at the incident wavelength of $10.653\mu\text{m}$, the difference between normal graphene plasmons and AGPs can be understood from the wavelength and back-gate voltage dependent dispersion of the AGPs and GPs mode (Fig.r14). It can be seen

AGPs have the following advantages compared with GPs:

- (a) the shorter wavelength under the same doping;
- (b) as the Fermi energy increases, the wavelength of the AGPs changes slowly - it is still at a very short wavelength, so the local field remains so strong;
- (c) the loss of the AGPs is smaller than that of GPs. In addition, the group speed of the AGPs will decrease as the gap decreases. [Ref. 15, Alonso-Gonzalez P., Nat Nanotechnol 1, 12 (2017)].

Of course, the realization of AGPs is more complicated than normal graphene plasmons.

Except for that, in the paper, one of the main conclusions is that AGPs show stronger scattering ability for extremely small steps than normal graphene, which can be used for the control of optical field in the atomic level.

Fig.r14: The wavelength and back gate voltage dependent dispersion of AGPs and GPs modes at the excitation wavelength of 10.653 μm .

In the revised version, we have added in Paragraph , “where U is back gate voltage between Au and graphene. Compared to traditional GPs, at the same gate voltages (same E_F), the AGPs has a shorter wavelength and a more localized field. The white circles on the color map are the experimental results,.....”

--- **Referee 3, point 5.** “None of these important questions was addressed or even touched yet. After reading the MS, the reviewer does not feel learn much new from it. They spend a length to explain that these fringes patterned are initiated by the non-uniform Au protrusions. But it is unclear what is new physics or potential implications of these studies.”

---Reply. Thank you for your comments. Because the source of the fringes is very important, so we spend a length to explain that these fringes patterned are initiated by the non-uniform Au protrusions. And from the main text (Fig.2), it can be known that, for a step with height of 3 nm, the reflectivity of AGPs is four times than GPs. Therefore, compared with GPs, AGPs can be used as: (1) atomic scale height sensor; (2) quantitative protein detection. The specific content can refer to **Referee 1, point 7** (Fig.r7).

On Page 7 of the conclusion part, we added: “Because of the unique properties that strong in-plane scattering of infrared light can happen even if the steps are close to atomic level, many applications can be envisioned. Firstly, the ultra-fine height sensors can be realized based on that the scattering amplitude of AGPs depends a lot on the height. Except for that, if the hole region can be filled or replaced with few layer protein or chemical molecules with infrared absorption bands, AGPs can be exploited in ultra-sensitive material sensing (Fig. S15 in supplementary). In addition, by controlling the parameters, strong scattering with multiple centers may enter the Anderson localization range, which could find applications in disorder physics.”

--- **Referee 3, point 6.** “It is also noticed that acoustic phonon studies in sandwiched graphene using the same technique and identical device structures have been reported in several other journals including Nature Comm itself. Thus it is hardly to be convinced that the current work is suitable for the next steps.”

---Reply. Thank you for your comments. Due to the unique properties of AGPs, many works have been explored in these field on both theoretic and experimental sides. However, the experiment work on AGPs is still few to the best of our knowledge. In the following, the related experimental works are listed in the following table.

	Work	Experimental method	novelty
Ref.1	Nat. Nanotech. 12, 31 (2017) (Hillenbrand group)	THz Near-field photocurrent nanoscopy	real-space imaging of acoustic THz plasmons in a graphene photodetector with split-gate architecture by nanoscale-resolved THz photocurrent near-field microscopy
Ref. 2	Nat. Nanotech. 14, 313(2019) (Sang-Hyun Oh group)	Mid-infrared FTIR	Realize a graphene acoustic plasmon resonator with nearly perfect absorption (94%) of incident mid-infrared light
Ref.3	Science 357, 189 (2017) (Koppens group)	THz near-field photocurrent nanoscopy	probe the nonlocal response of the graphene electron liquid by acoustic plasmons
Ref.4	Science 368, 1219 (2020) (Koppens group)	Mid-infrared FTIR	Single acoustic plasmon cavity with ultracompressed mode volumes
Ref.5	Science 360, 291	Mid-infrared FTIR	plasmon confinement

	(2018) (Koppens group)		down to the ultimate limit of the length scale of one atom
Ref.6	Nat. Commun. 12, 938 (2021) (Min Seok Jang group)	Mid-infrared near-field optical microscopy	acoustic plasmons in large-area graphene grown by chemical vapor deposition
	Our work	Mid-infrared near-field optical microscopy	Strong scattering of acoustic plasmon by atomic steps

As we can see, Koppens' group indeed have done many excellent works in the field of acoustic graphene plasmons (Ref.3-5), they have shown isolated acoustic plasmon cavities with extremely field confinement and used acoustic graphene plasmon to probe the extremely field localization and quantum effect of electrons. However, far-field optical spectrum or near-field photoncurrent nanoscopy is used, which are obviously different from our work.

Moreover, Ref1 from Hillenbrand's group have demonstrated that near-field photocurrent nanoscopy can be used for polariton exploration at THz range. Ref2 provides a two-step coupling mechanism to realize over 90% absorption of infrared light and proves that acoustic plasmons can be used for ultrathin protein analysis. Ref.6, which is published after our submission, the only work which takes the same experimental tools as us, but the major novelty of the paper is that large area CVD graphene shows good acoustic plasmon response as well compared with mechanical exfoliated graphene.

Though the study of AGPs have been mentioned, we would like to stress that unlike the impression the Referee may have had, our manuscript is contributing to the in-plane scattering of AGPs, which has never been mentioned, and it is essential for the manipulation and utilization of ultraconfined optical field down to atomic level. As a result, we believe our work should be competitive for Nat. Commun., and hope he/she will consequently reconsider the recommendation.

In the revised version, we have stressed more on our novelty compared with other work in paragraph 2 in the Introduction part as "Yet, though lots of work on AGPs have been reported, the experimental progress is still few. Specifically, both near-field photocurrent nonoscopy [15,17] and far-field infrared spectrum [12, 16, 18] are used to explore the properties of AGPs in either THz or infrared ranges, however, the direct detection of AGPs through infrared near-field optical scattering [21] is still difficult, especial on the in-plane scattering properties of AGPs.. And this is essential for the manipulation and utilization of ultraconfined optical field down to atomic level."

Best regards

Wei Cai

REVIEWER COMMENTS

Reviewer #1 (Remarks to the Author):

I thank the authors for addressing my comments in the revision, especially via the supplementary data and analysis. The current manuscript can be considered for publication in Nature Communication.

Reviewer #2 (Remarks to the Author):

I have read the response to the referees and the revised manuscript. The authors have clarified various points that were unclear in the manuscript, and emphasized that the novelty lies in the observation of strong in-plane scattering.

Still, I am not convinced that this work merits publication in Nature Communications. The direct probe of in-plane scattering from a defect is nice, but the scattering is largely to be expected.

There are some theoretical arguments to expect that AGPs scatter more strongly than GPs, but it is not terribly clear what the implications of this are. Also, it is not clear if the comparison is fair, as two different dispersion relations are being compared at the same wavelength, but this does not translate to an intrinsic improvement of scattering. For example, it appears for larger frequencies that the scattering length of GPs becomes favorable to AGPs, so some more analysis could be done to show how the maximal scattering compares between GPs and AGPs. The authors assert that it is crucial for control of light at the nanoscale, but give little justification. I am ultimately concerned that it is not clear how these results could translate into impact for other studies of AGPs.

If the authors provided some theoretical justification for how their results imply strong light-matter interactions with AGPs generally (that could be useful in many other studies besides scattering from defects), that would help their case. The analysis of sensing provided in response to Referee 1 is somewhat interesting along these lines: I have some questions about how this depends on wavelength (similar to my comment about the dependence of scattering length on wavelength), but this could be an interesting example of how AGPs could be uniquely useful.

Reviewer #3 (Remarks to the Author):

I have reviewed the revised version of the Acoustic graphene plasmons work done by Wei Cai and his group. The authors had tried very hard to answer most of my concerns and questions. With that, I recommend to accept this work.

Response to Reviewer #1

“I thanks the authors for addressing my comments in the revision, especially via the supplementary data and analysis. The current manuscript can be considered for publication in Nature Communication.”

---Reply. We are very grateful to the reviewer for providing so many valuable concerns on our manuscript, which have helped us improve the quality of the manuscript.

Response to Reviewer #2

“I have read the response to the referees and the revised manuscript. The authors have clarified various points that were unclear in the manuscript, and emphasized that the novelty lies in the observation of strong in-plane scattering.

--- Reply. We thank the reviewer a lot for affirming our efforts in improving the manuscript.

--Referee2, Point 1. “Still, I am not convinced that this work merits publication in Nature Communications. The direct probe of in-plane scattering from a defect is nice, but the scattering is largely to be expected.

There are some theoretical arguments to expect that AGPs scatter more strongly than GPs, but it is not terribly clear what the implications of this are.”

---Reply. We agree with the referee on the concern. Except for the demonstration of strong light scattering by atomic steps, explaining the applications of strongly scattered AGPs on photon-matter interactions is important as well for stimulating related research on this field. In fact, although light scattering is an old physical problem, which can be traced back to Isaac Newton in the 17th century, it still raises wide attention recently because these effects can be explored at the nanoscale. For example, lab-on-a-chip spectroscopy applications [Nature Photonics 7, 746 (2013)], subwavelength topological phases [Nature communications 8, 16023 (2017)] and strong directional scattering with broadband spectral range [Nature communications 3, 692 (2012)] have been demonstrated based on light scattering at the nanoscale. Specially, in the vertical dimension, Anderson localization based on the interference of coherent backscattering waves in deep subwavelength multilayer has been observed [Science 356, 953, (2017)], though the disorder in each layer is only 2-nm. In the in-plane dimension, strong reflection of plasmon waves by nm height step of discontinuous graphene is demonstrated [Nano letters 13, 6210 (2013)]. Naturally, a fundamental question arises, for a fixed frequency, how large scatterer size needed for obvious scattering effect? And by reducing this limit to atomic scale, optical applications at atomic scale can be expected. Just as pointed out in the Viewpoint by the Nobel laureate Prof. A. K. Geim [Nano Lett, 21, 6356 (2021)], because of the advances in 2D Van der waals materials, the electric and chemical explorations and applications of 2D cavities with atomic height are boosting. We believe that studies of nanoscale structure shown in our manuscript can shed new light on this field. Our work is a preliminary study on this field. Although only single scattering event is explored, multi-scattering, coherent scattering and even Anderson localization effect can be expected in this system.

Except for that, as suggested by the referee and our first-round replies to Ref.1 and Ref. 3, our studied scheme can be potentially used for ultra-thin chemical layer sensing. For example, when encountering defects or changes of the dielectric environment, AGPs can be clearly detected compared to GPs. This has been illustrated as following.

Fig.r1: The absorption spectra of the AGPs and GPs when pass through protein molecules of different heights at 0.5, 1.0 and 1.5 nm.

Here, the analytic model is used to retrieve the protein permittivity from experimental results by adjusting a Lorentzian permittivity:

$$\varepsilon_p(\omega) = n_\infty^2 + \frac{s_1^2}{\omega_1^2 - \omega^2 - i\omega\gamma_1} + \frac{s_2^2}{\omega_2^2 - \omega^2 - i\omega\gamma_2}, \quad n_\infty^2 = 2.08,$$

$$\omega_1 = 1668\text{cm}^{-1}, \omega_2 = 1532\text{cm}^{-1}, \gamma_1 = 78.1\text{cm}^{-1}, \gamma_2 = 101\text{cm}^{-1}, s_1 = 213\text{cm}^{-1}, s_2 = 200\text{cm}^{-1}$$

Fig.r1(a) and (b) shows that when AGPs and GPs pass through a nano-scale (the height of protein is 0.5 nm, 1.0 nm and 1.5 nm) protein molecule (pyridine), AGPs will form very strong transmittance peaks at the two natural frequencies of the protein molecule, but the GPs will only form absorption peaks at ω_1 under the same circumstances. The transmittance of AGPs is significantly larger than that of GPs. It shows that AGPs have great advantages in biosensing of nano-scale protein compared to GPs.

On Page 2 of the main text, we have added: “And this is essential for the manipulation and utilization of ultraconfined optical fields down to atomic level, where the widely applications of two-dimensional atomic height cavities are booming thanks to the advancement of two-dimensional van der Waals materials [22].”

On Page 7 of the main text, in the second paragraph of the conclusion, we have added: “Moreover, the individual scatterers show the excellent near-field response of AGPs, which can be efficiently tuned by the gate voltage. Our work is a preliminary optical study on 2D cavities with atomic height [22], although only single scattering event is explored, multi-scattering, coherent scattering and even Anderson localization effect [41] can be expected in this system.”

--Referee2, Point 2.” Also, it is not clear if the comparison is fair, as two different

dispersion relations are being compared at the same wavelength, but this does not translate to an intrinsic improvement of scattering.”

---Reply. The scattering effect depends a lot on dimensional ratio between the source and scatterer. We agree with the referee that GPs and AGPs possess different plasmon wavelengths due to the different dispersion. The advantage of AGPs is that they have much shorter wavelength and stronger field localization than GPs. We understand the referee that it seems fairer to compare them at the same plasmon wavelength, but in this case, they should have different frequencies. In practical applications, different physical phenomena, like bio-sensing and integrated photonics circuit are advanced at specific frequencies. Therefore, comparing physical effects under the same incident source may be more convincing.

--Referee2, Point 3. “For example, it appears for larger frequencies that the scattering length of GPs becomes favorable to AGPs, so some more analysis could be done to show how the maximal scattering compares between GPs and AGPs. The authors assert that it is crucial for control of light at the nanoscale, but give little justification. I am ultimately concerned that it is not clear how these results could translate into impact for other studies of AGPs. ”

---Reply. We apologize for this unclear explanation in previous reply. As seen in Fig. 2(a) and the first-round reply, it seems that the scattering length of GPs can exceed that for AGPs in some regions. However, this effect comes from the phonon absorptions in the interlayer (hBN), which means that AGPs cannot be well formed. To illustrate this question, dispersion relation of Gra/hBN(6 nm)/Au(50 nm) and Gra/hBN(6 nm)/SiO₂(500 nm) were simulated and shown as Fig.2.(b)-(c). One can find that the frequency region where the scattering length of GPs over that of AGPs is at the position of the hBN phonon, which form band gaps in the dispersion plots. And this affects more on AGPs than GPs because more intense electric fields exist in the hBN layer for AGPs, where the interaction energy can be approximately described as $-p \cdot E$. Moreover, outside the phonon induced bandgap region, AGPs always possess shorter wavelength than that of GPs. In other words, by avoiding the phonon region of the interlayer hBN or replacing with another materials without phonon response, the scattering of AGPs will be always larger than that of GPs.

Fig.r2: The relation between scattering length and dispersion of AGPs and GPs. (a) The scattering length of AGPs and GPs by a hole-like structure. (b)The dispersion relation of Gra/BN(6 nm)/Au(50 nm); (b) The dispersion relation of Gra/BN(6 nm)/SiO₂(500 nm).

On Page 12 of supporting information, in FIG. S10, we have added the dispersion relation diagram (c) and (d) and the description “(c) The dispersion relation of Gra/BN (6 nm)/Au (50 nm); (d) The dispersion relation of Gra/BN (6 nm)/SiO₂ (500 nm).” .

On Page 12 of the supporting information text, line 4, we have added: “The valley where the scattering length of GPs over that of AGPs corresponds to the position of the hBN phonons. To illustrate this point, dispersion relation of Gra/hBN (6 nm)/Au (50 nm) and Gra/hBN (6 nm)/SiO₂ (500 nm) were simulated and shown in FIG. S10(c) and FIG. S10(d). From that, it can be known the valley just falls in the forbidden zone of the dispersion relation.”

--Referee2, Point 4. “If the authors provided some theoretical justification for how their results imply strong light-matter interactions with AGPs generally (that could be useful in many other studies besides scattering from defects), that would help their case. The analysis of sensing provided in response to Referee 1 is somewhat interesting along these lines: I have some questions about how this depends on wavelength (similar to my comment about the dependence of scattering length on wavelength), but this could be an interesting example of how AGPs could be uniquely useful.”

--- Reply. Thanks a lot for the kind suggestion, we have supplemented the ultra-thin layer chemical sensing application in the revised version, which clearly shows that AGPs can imply stronger light-matter interactions than GPs. Moreover, as aforementioned, the ability of sensing for AGPs is always greater than that of GPs, as long as the work wavelength is outside the phonon absorption region of the interlayer hBN layer, providing the well-formed AGPs. We have pointed out this important prerequisite for the real AGPs applications.

Except for the sensing application, we believe our work would stimulate light scattering studies at the atomic scales, such as Anderson localization and other transport phenomena, promoting applications in imaging and random lasing [Nature Photonics 7, 188 (2013)]. In addition, our scheme may find applications in resolving intrinsically quantum mechanical effects with subnanometer resolution [Nat Commun 12, 3271 (2021)].

On Page 7 of the conclusion part, we have added: “Except for that, if the hole region can be filled or replaced with few layer protein or chemical molecules with infrared absorption bands, AGPs always show better ultra sensitive material sensing than GPs, as long as the phonon frequency region of the interlayer hBN is avoided (Fig. S15 in supplementary). Furthermore, our scheme may find applications in resolving intrinsically quantum mechanical effects with subnanometer resolution [40].”

On Page 7 of the main text, in the second paragraph of the conclusion, we have added: “Moreover, the individual scatterers show the excellent near-field response of AGPs, which can be efficiently tuned by the gate voltage. Our work is a preliminary optical study on 2D cavities with atomic height [22], although only single scattering event is

explored, multi-scattering, coherent scattering and even Anderson localization effect [41] can be expected in this system.”

Response to Reviewer #3

“I have reviewed the revised version of the Acoustic graphene plasmons work done by Wei Cai and his group. The authors had tried very hard to answer most of my concerns and questions. With that, I recommend to accept this work.”

---Reply. We are very grateful to the reviewer for providing many valuable concerns on our manuscript, which have helped us improve the quality of the manuscript.

REVIEWER COMMENTS

Reviewer #2 (Remarks to the Author):

I have read the revised manuscript and the response to all comments from the Referees; the manuscript may now be accepted.